# A Homeostasis Hypothesis of Avian Influenza Resistance in Chickens

**DOI:** 10.3390/genes10070543

**Published:** 2019-07-17

**Authors:** Jing An, Jinxiu Li, Ying Wang, Jing Wang, Qinghe Li, Huaijun Zhou, Xiaoxiang Hu, Yiqiang Zhao, Ning Li

**Affiliations:** 1Beijing Advanced Innovation Center for Food Nutrition and Human Health, College of Biological Sciences, China Agricultural University, Beijing 100193, China; 2State Key Laboratory of Agrobiotechnology, College of Biological Sciences, China Agricultural University, Beijing 100193, China; 3Department of Animal Science, University of California, Davis, CA 95616, USA

**Keywords:** Fayoumi, Leghorn, difference degrees, fold change levels, methylation, F_ST_, RNA-seq

## Abstract

Avian influenza has caused significant damage to the poultry industry globally. Consequently, efforts have been made to elucidate the disease mechanisms as well as the mechanisms of disease resistance. Here, by investigating two chicken breeds with distinct responses to avian influenza virus (AIV), Leghorn GB2 and Fayoumi M43, we compared their genome, methylation, and transcriptome differences. *MX1*, *HSP90AB1*, and *HSP90B1* exhibited high degrees of genetic differentiation (F_ST_) between the two species. Except for the *MX1*-involved direct anti-virus mechanism, we found that at the methylation and transcriptome levels, the more AIV-resistant breed, Fayoumi, exhibited less variation compared with Leghorn after AIV inoculation, which included change trends in differentially expressed regions, top-fold change genes with FDR-corrected *p* < 0.05, immune response related genes, and housekeeping genes. Fayoumi also showed better consistency regarding changes in methylation and changes at the transcriptome level. Our results suggest a homeostasis hypothesis for avian influenza resistance, with Fayoumi maintaining superior homeostasis at both the epigenetic and gene expression levels. Three candidate genes—*MX1*, *HSP90AB1*, and *HSP90B1*—showed genetic differentiation and altered gene expression, methylation, and protein expression, which merit attention in further functional studies.

## 1. Introduction

Avian influenza is a highly contagious disease. The disease is caused by avian influenza virus (AIV), which is classified as belonging to the Orthomyxoviridae family [1]. Avian influenza virus has not only caused great economic damage, but also poses a serious threat to human lives [2]. Numerous AIV vaccines have been developed to prevent its spread in chickens or from chickens to humans. However, due to the rapid mutation of the pathogens and time taken to develop effective vaccines, AIV has not been under complete control. Consequently, understanding the mechanisms of AIV inoculation and resistance is urgent to better control avian influenza outbreaks.

Different bird species exhibit distinct immunologic responses towards AIV infections [3]. It is reported that *MX1* is relevant to reductions in morbidity, viral shedding, and cytokine responses in chickens with highly pathogenic AIV inoculation [4,5]. Researchers created genetically modified chickens, which overexpressed a small hairpin RNA with a chicken U6 promoter that regulated viral RNA polymerase, making them unable to transmit AIV to other chickens [6]. Transgenic chickens expressing the 3D8 single chain variable fragment protein have also been generated, which was demonstrated to suppress avian influenza transmission [7].

Due to its superior reproduction trait, the Leghorn chicken plays an important role in the commercial egg supply market [8]. Unfortunately, however, it has very little ability to resist diseases, including avian influenza. Fayoumi is an indigenous chicken breed that originated in Egypt. It is well known for its strong resistance to various pathogens, and it is very robust to extreme environmental challenges [9,10]. To discover the underlying mechanisms of distinct phenotypic differences in avian influenza resistance, Leghorn GB2 and Fayoumi M43 have previously been compared at the DNA, expression, and methylation levels [9,11,12,13]. Wang et al. identified differentially expressed genes before and after the avian influenza virus inoculation and reported novel signaling pathways associated with disease resistance, including oxygen transport, gas transport, and the establishment of localization [12]. Recently, Li et al. compared differences in the whole genome methylation pattern between the two breeds [11]. However, the methylation data were collected under normal conditions only for the two breeds, and no AIV inoculation experiments were performed. It is worth noting that disease resistance does not mean free from infection. A previous study showed a considerable amount of virus titers in infected Fayoumi, even though they appeared less clinically ill than the Leghorn chickens [12]. This suggests that they have an alternative approach to disease resistance, whereby they have better survival rates after contracting the virus, in addition to having a direct anti-virus mechanism.

Homeostasis is defined as the ability to maintain a stable internal environment in the presence of disturbances [14], which has been observed in numerous biological processes [11,15,16,17,18,19], such as oxidative stress and calcium metabolism [20]. After AIV invades the hosts, besides activating anti-virus procedures including the NF-κB signaling pathway or the RIG-I signaling pathway, it is also important to protect hosts from tissue lesions that result from immune over-reaction, i.e., surviving with reduced body damage. This is consistent with the concept of canalization that was originally described by Waddington [21], which predicts the ability to maintain a constant phenotype regardless of environmental perturbations [22]. In this case, AIV is regarded as the environmental perturbation.

In this study, we extended our previous work on the DNA methylation comparison between Leghorn and Fayoumi chickens [11]. We investigated differentially methylated regions as well as top-fold change genes with FDR-corrected *p* < 0.05, between individuals with AIV (avian influenza H5N3, low pathogenicity avian influenza, LPAI) inoculation compared to individuals challenged with PBS in both breeds. We found that for both methylation and transcriptome levels, the more AIV-resistant breed, Fayoumi, exhibited less variation compared to Leghorn chickens after AIV inoculation. Based on our results, we proposed a novel hypothesis for an AIV resistance mechanism, which could be used as a supplement to traditional anti-virus mechanisms that focus on destroying or blocking the virus. Namely, Fayoumi can better maintain internal homeostasis in both methylation and gene expression levels to avoid over-reactions due to AIV infections.

## 2. Materials and Methods 

### 2.1. Samples and Experimental Design

All experiments were conducted in two genetically distinct and highly inbred chicken lines, Leghorn GB2 and Fayoumi M43, which have an inbreeding coefficient of more than 99.9% and the specific experimental designs has been listed as follows. The samples and replicates have been artificially selected by the Iowa State group since 1954 [13]. Blood DNA samples from 16 individuals per line were pooled for resequencing [13]. Whole genome wide methylome samples were generated from eight lung tissue libraries for each line, which were inoculated with avian influenza H5N3 (LPAI) or PBS (Phosphate Buffer Saline), and lung samples were harvested at 96 h post-inoculation. RNA sequencing samples were generated from the same experimental design and generated from Wang et al. [12]. All animal experiments were conducted in our collaborative laboratory at the Texas A&M University, and data has been published previously [12]. 

### 2.2. RNA-seq Data Analysis

The samples used for RNA-seq data analysis were taken from Wang et al. [12]. Eight three-week-old chickens were randomly chosen from individuals challenged with a 10^7^ 50% egg infective dose of H5N3 AIV (LPAI) or PBS. There were two birds per breed per treatment. Based on a previous test, which showed that lesion severity in chicken lungs hit its peak at four days post-inoculation, the total RNA of the lungs and trachea was harvested and extracted by the trizol method in accordance with the manufacturer’s instructions [12]. Reads with adapters and low Phred qualities were also removed by the FASTX-Toolkit (http://hannonlab.cshl.edu/fastx-toolkit). Qualified reads were mapped against the reference genome (GRCg6a) using the Tophat software [23] with default parameters. Bam files were utilized to assemble transcripts using cufflinks [23]. Top-fold change genes with FDR-corrected *p* < 0.05 were identified by cuffdiff [23]. The gene ontology (GO) term annotation was conducted using the bioconductor topGO package, which used Fisher’s exact test to perform enrichment tests of top-fold change genes. In addition, to reveal the enrichment degrees of relevant pathways in both lines, bubble plots were plotted using the ggplot2 package, and these are presented in Appendix A.

Top-fold change genes were calculated using cuffdiff, with the level of significance set as an FDR-corrected *p*-value below 0.05. The gene expression difference and fold change levels of the top-fold change gene set, the immune response related gene set, the housekeeping gene set, and the gene set as a whole were calculated as follows: (1) Fayoumi gene expression difference degrees were measured as the mean values of log_2_(Fayoumi gene expression values with AIV inoculation/Fayoumi gene expression values challenged with PBS) (log_2_(FA_G/FP_G)) and Fayoumi gene expression fold change levels were measured as the variance of log_2_(FA_G/FP_G); (2) Leghorn gene expression difference degrees were measured as the mean value of log_2_(Leghorn gene expression values with AIV inoculation/Leghorn gene expression values challenged with PBS) (log_2_(LA_G/LP_G)) and Leghorn gene expression fold change levels were measured as the variance of log_2_(LA_G/LP_G); (3) a comparison of gene difference between Fayoumi and Leghorn was conducted using t-tests and a comparison of gene expression fold change levels between Fayoumi and Leghorn was conducted using F-tests. 

### 2.3. Whole Genome Bisulfite Sequencing Data Analysis

DNA samples of lung tissue were collected at 96 h post-inoculation with avian influenza H5N3 (LPAI) or PBS, respectively. These DNA samples were extracted according to the phenol chloroform isoamylalcohol method. After DNA was sonicated to 200–400 bp, an end repair job was completed with a triphosphate mix free of dCTP. The following ligation procedures, methylation adapters, and DNA library sequencing were conducted on the Illumina sequencing platform. Raw whole genome methylation sequencing data were generated from the Illumina Genome Analyzer II. The processed reads were then converted and mapped against the reference genome (GRCg6a) using the Bismark software [24], which was also used to quantify the methylation ratios of single sites in the chicken genome. Differentially methylated sites and differentially methylated regions were calculated by Radmeth [25] under the beta-binomial regression model with an FDR-corrected *p*-value below 0.01. 

The DMRs locations were annotated with the reference annotation file (GRCg6a gtf). The gene body was defined as the region from the start codon to the stop codon. The promoter region was defined as 5 kb upstream of the start codon, while the upstream region was defined as 5 kb upstream of the promoter, and the distant region was defined as 5 kb upstream of the upstream region. The methylation level of each gene was obtained by averaging the methylation ratios of individual sites in the gene body. Since four replicates were inoculated with AIV or challenged with PBS in both Fayoumi and Leghorn chickens, log_2_(gene methylation ratios of samples with AIV/gene methylation ratios of samples with PBS) were calculated for each individual gene or particular gene set to determine the methylation difference as well as the fold change level. Again, t-tests were used to compare the gene methylation difference, and F-tests were used to compare gene methylation fold change levels between Fayoumi and Leghorn chickens. 

### 2.4. Resequencing Data Analysis

Genomic resequencing data from the Fayoumi and Leghorn breeds were retrieved from Fleming et al. [13]. Raw reads were mapped against the reference genome (GRCg6a Ensemble database version) using BWA (Burrows-Wheeler Alignment tool) [26] with default settings. The Genome Analysis Toolkit (GATK) 3.2.2 [27] was utilized for SNP calling for each group following the GATK best practice pipeline. The fixation index (F_ST_) values at single nucleotide resolution were calculated according to the Weir and Cocker-ham (1984) formula [28] using in-house Perl scripts. By employing a sliding window approach with a window size of 10 Kbp and a step size of 10 Kbp, the genetic differentiation between Fayoumi and Leghorn was averaged for each window throughout the whole genome. Similarly, to quantify genetic differentiation at the gene level, F_ST_ values for individual genes were obtained by averaging single nucleotide F_ST_ values from the start codon to the stop codon of a gene.

### 2.5. dn/ds Ratio Calculation

To calculate the dn/ds ratio (the ratio of the number of synonymous substitutions per synonymous site to the number of nonsynonymous substitutions per nonsynonymous site) for *MX1*, *HSP90AB1,* and *HSP90B1*, we first extracted CDS sequences for the reference genome, as well as for Fayoumi and Leghorn, using the bedtools getfasta function [29]. Then, fasta sequences were translated into amino acids with the HyPhy 2.2.4 version data file tools [30]. The dn/ds ratios of the three genes were measured on the Datamonkey 2.0 server [31], which was based on the mixed effects branch-site model of the MEME software [32].

### 2.6. Data Availability

Our sequencing data have already been submitted to the NCBI Gene Expression Omnibus (http://www.ncbi.nlm.nih.gov/geo) under accession nos. GSE56975 and GSE128053.

## 3. Results

### 3.1. Less Gene Expression Fold Change Levels in Response to Avian Influenza Virus in Fayoumi

Since our RNA-seq data were generated from Wang et al. [12], the phenotype differences in AIV resistance between Fayoumi and Leghorn have already been tested. There were two birds per breed per treatment. Infected chicken lungs hit the peak at four days post-inoculation, and virus titers in infected Fayoumi 5.270 log_10_ EID_50_ mL^−1^ (EID_50_ = 50% egg infective dose) were lower than those in Leghorn chickens (6.480 log_10_ EID_50_ mL^−1^). The lesion scores for Fayoumi and Leghorn chickens were 55% and 80%, respectively [12], which indicated that Fayoumi chickens can block viral entry/replication to some degree and were healthier compared to Leghorn chickens after AIV inoculation. To explore the altered expression patterns in both breeds, we re-analyzed the RNA-sequencing data. 

The results showed that after being inoculated with AIV, many genes exhibited altered expression patterns. Thirty-five differentially expressed genes (*p* < 0.05 adjusted by FDR) were mainly enriched in the pathways of respiratory burst and mitochondrial calcium ion homeostasis between Fayoumi individuals with AIV inoculation (FA) compared to Fayoumi individuals challenged with PBS (FP) FA compared to FP (Appendix A). Fifty differentially expressed genes (*p* < 0.05 adjusted by FDR) were mainly enriched in the pathways of cell metabolism and response to xenobiotic stimulus between Leghorn individuals with AIV inoculation (LA) compared to Leghorn individuals challenged with PBS (LP) (Appendix A). For both Leghorn and Fayoumi chickens, differentially expressed genes were enriched in function terms related to the immune response (Appendix A).

For the top-fold change gene set, the fold change or difference between expression levels after inoculation with AIV was calculated for both Leghorn and Fayoumi (see Materials and Methods). The comparison using t-tests showed that the difference between expression levels of Fayoumi was significantly smaller than those of Leghorn chickens, with a *p*-value of 0.026. As shown in Figure 1A, the F-test results revealed that the gene expression fold change levels of top-fold change genes in Fayoumi were also significantly smaller than those in Leghorn chickens with a *p*-value of 0.049. This suggests that Fayoumi can better maintain internal homeostasis following exposure to an external stimulus, e.g., AIV inoculation, while Leghorn chickens are more susceptible to the pathogen and have more difficulty stabilizing gene expression.

For immune response related genes (genes listed in Appendix A), by comparing log_2_(FA_G/FP_G) and log_2_(LA_G/LP_G), more genes were observed to be upregulated in innate immune responses in Leghorn chickens, whereas more genes were upregulated in the adaptive immune response in Fayoumi chickens (as shown in Appendix A). In this case, the immune response may constitute a double-edged sword, in which it can both inhibit replications of AIV and damage the organism itself. We also noticed fold changes of genes involved in tissue lesions, such as those involved in the oxidative stress pathway, the apoptosis pathway, and the ferroptosis pathway, were much smaller in Fayoumi compared with Leghorn chickens (Appendix A). Interestingly, it was found that the gene expression change degrees of the homeostasis indicator *HSP90AB1* [33] were upregulated more in Fayoumi compared to in Leghorn chickens (log_2_(FA_G/FP_G)/log_2_(LA_G/LP_G))= 6.0851, Appendix A). As a supplement to traditional anti-virus mechanisms that focus on destroying or blocking the virus, Fayoumi chickens may be able to better maintain internal homeostasis in gene expression levels allowing them to avoid over-reactions due to AIV infections.

### 3.2. Avian Influenza Virus Inoculation Introduced Changes in DNA Methylation in Both Breeds

To reflect dynamic responses to an external stimulus, we performed whole genome bisulfite sequencing (WGBS) on the lung tissues of two three-week-old chickens (Fayoumi and Leghorn) that were either inoculated with 10^7^ 50% egg infective dose (EID_50_) of low-path H5N3 AIV or challenged with PBS. For each individual, we generated two technical replicates. In total, 16 WGBS libraries were generated (two breeds × two conditions × two individuals × two replicates) (detailed in Materials and Methods). Filtered clean reads from the 16 WGBS sequencing data ranged from 154 million to 516 million per sample (Appendix A). The mapping rates were about 86.2% (Appendix A).

The majority of CpG sites (60~80%) in the genome were methylated, which is consistent with a previous study [34]. As shown in Appendix A, we plotted the average DNA methylation level of all 16 samples proportionately along the gene structure. To reveal the difference in the methylation pattern, we focused on two comparisons: FA compared to FP and LA compared to LP. For each comparison, analyses of differentially methylated sites (DMSs) and differentially methylated regions (DMRs) were performed using the Radmeth software [25]. As listed in Appendix A, we identified 2143 DMSs from the FA compared to FP comparison and 162 DMSs from the LA compared to LP comparison. It is well known that DNA methylation functions differently depending on its location [35]. Methylation in the gene body has been reported to be relevant to the process of alternative splicing [36], while methylation in promoters is associated with transcriptional activation [35]. To explore the distribution features of DMRs, we scanned the locations of DMRs for both breeds (FA compared to FP and LA compared to LP). In the gene region, 10.5% of DMRs were located in the gene body, and the rest were annotated at the promoter and distant regions (>10 kb). Within the gene body, 29.4% of DMRs were located at the lncRNA (long non-coding RNA) genes. To test whether the differences in methylation sites or degrees between the two breeds were due to unequal efficiency of methylation-related enzymes, we checked the changes in gene expression for methylation-related enzymes. For FA compared to FP and for LA compared to LP, DNMT1 was downregulated with fold changes of 0.9215 (*p* = 0.8663) and 0.6433 (*p* = 0.4283), respectively. For DNMT3a, the fold change in FA compared to FP was 1.0605 with *p* = 0.7953, while for the Leghorn group, the fold change was 1.5638 with *p* = 0.1983. For DNMT3b, the corresponding values were 0.5433 and 1.0378 for Fayoumi and Leghorn chickens, respectively (*p* > 0.05).

### 3.3. Less DNA Methylation Fold Change Levels in Response to Avian Influenza Virus in Fayoumi

In addition to identifying DMSs and DMRs, the fold changes or differences in methylation levels of DMRs were also measured (see Materials and Methods). The comparison using t-tests showed that the Fayoumi methylation difference level was much smaller than those of Leghorn chickens with *p* < 0.0001. As shown in Figure 1B, the methylation fold change levels in Fayoumi chickens were also significantly smaller than those in Leghorn chickens (F-test, *p* < 0.0001), which again supported the hypothesis that Fayoumi chickens can better maintain internal homeostasis. To explore the functional roles of DMRs in Fayoumi and Leghorn chickens, we annotated genes located at DMRs, as listed in Appendix A. We found that multiple genes were enriched in immune response related pathways. 

We separately addressed the methylation status of immune response related genes (the same gene lists as in Appendix A: Innate immune response, adaptive immune response, and oxidative stress related pathways). To retrieve the gene level methylation ratios, we first averaged methylation ratios of each site along a gene from the start codon to the stop codon. For all immune response related genes, we calculated the methylation change degree. We found it was smaller in Fayoumi compared to Leghorn chickens, although it was not shown to be statistically significant using t-tests (Appendix A). However, as shown in Figure 2, the gene methylation fold change degree of most immune response related genes was significantly smaller in Fayoumi than in Leghorn chickens. This suggests that Fayoumi chickens have a greater ability to avoid an excessive immune response.

### 3.4. Further Tests for Homeostasis Hypothesis in Fayoumi and Leghorn Breeds

In Fayoumi chickens, the overall smaller changes in the difference or fold change levels in methylation or expression, with milder symptoms after AIV inoculation, supported the hypothesis of maintenance of homeostasis for disease resistance. To further test this hypothesis, we calculated the Pearson coefficient between difference levels of DMRs and difference levels of corresponding top-fold change genes with an FDR-corrected *p* < 0.05. For Fayoumi chickens, a positive correlation of 0.370 was observed (*p* = 0.030), whereas a significant correlation was not found for Leghorn chickens (r = 0.119, *p* = 0.337). This result suggested more efficient epigenetic regulation of gene expression in Fayoumi chickens compared with Leghorn chickens. 

Considering that housekeeping genes are generally more stable, if the homeostasis hypothesis holds true, we expect distinct difference and fold change levels in housekeeping genes between the two breeds. As shown in Figure 3, the distribution of Fayoumi housekeeping gene expression difference was more contracted compared to the distribution in Leghorn chickens. A paired t-test was utilized to compare the mean difference in both breeds, and the housekeeping gene expression difference levels in Leghorn were significantly larger (*p* = 0.001) than those in Fayoumi, with t = −3.2307. F-tests were also used to compare the expression fold change levels of housekeeping genes (variance of log_2_(IA_G/IP_G)) in both breeds, and the expression fold change levels of housekeeping genes in Fayoumi were observed to be much smaller than those in Leghorn chickens, with F = 1.119 and *p* = 0.002.

Methylation difference degrees and fold change levels were also calculated for all housekeeping genes. After averaging four replicates for each gene, we compared the methylation difference degrees of housekeeping genes in both breeds, and we found that the methylation difference degrees of Fayoumi housekeeping genes were much smaller than those of Leghorn housekeeping genes (t = −23.870, *p* < 0.0001). Methylation fold change levels of housekeeping genes showed statistical differences where Fayoumi chickens had much smaller levels than Leghorn chickens (F = 1.707, *p* < 0.0001), which is consistent with our hypothesis. In summary, these results suggest a distinct ability to maintain homeostasis in both breeds. 

### 3.5. Genetic Differentiation Between Fayoumi and Leghorn Chickens

The Fayoumi and Leghorn chickens used in this study are two highly inbred breeds with inbreeding coefficients of over 99.9% [9,11,13]. To elucidate the genetic differentiation between these two breeds at the whole genome level, we used the fixation index (F_ST_). By using a window size of 10 kbp and a step size of 10 kbp, we calculated the F_ST_ in every window and acquired its distribution for each chromosome. Consistent with our expectations, we observed that, for almost all chromosomes, the median F_ST_ value was over 0.900, except for chromosomes 6, 7, 8, 9, 16, 25, and 28 (Appendix A). Chromosome 16, where the MHC genes are located, showed a lower F_ST_ between Leghorn and Fayoumi chickens compared to most of the other chromosomes. We specifically investigated the genomic regions harboring MHC reported by Abernathy et al. [9]. The median F_ST_ values in the five MHC regions (195000–205000, 215000–225000, 95000–105000, 125000–135000, and 145000–155000 bp) were found to be 0.811, 0.743, 0.741, 0.714, and 0.707, respectively, which were also lower than the genome average value. These results indicated that the MHC-involved process of antigen presentation might not adequately explain the distinct resistance abilities between the two breeds. 

Furthermore, we calculated F_ST_ values for all 19542 genes in the chicken genome (downloaded from the Ensemble database GRCg6a) by averaging F_ST_ values of each base along the gene regions. The mean F_ST_ value for all genes was 0.564, and the 95% quantile value was 0.773. As shown in Figure 4, we surveyed immune response related genes by grouping them according to the order of immune response occurrence: The innate immune response, the adaptive immune response, and oxidative stress related pathways. *MX1*, *HSP90AB1*, and *HSP90B1* were the only three genes that exhibited high F_ST_ values (0.802, 0.787, and 0.972, respectively) in the top 5% of all genes. According to the gene annotations, *MX1* is involved in the innate immune response to AIV [4,5]. On the other hand, HSP90 family genes are known to be essential for protecting against the heat shock response and providing stress tolerance [33] by buffering proteostasis against environmental stress and functioning as a hub gene regulating protein homeostasis.

## 4. Discussion

Highly contagious avian influenza, which spreads through the respiratory tract [37], constitutes a serious burden for both the poultry industry and public safety [1]. The rapid development of the NGS (next generation sequencing) technology has provided a fruitful opportunity to investigate the mechanisms of disease resistance of avian influenza from a broader perspective. Here, we extended our previous research [11] by studying DNA resequencing, methylation, and expression data using two chicken breeds with extreme AIV resistance phenotypes. Our results suggest a homeostasis hypothesis for avian influenza resistance in chickens (Figure 5).

After AIV inoculation, survival requires a delicate balance between the host defense and host tolerance. The host defense is essential for the detection and elimination of foreign pathogens, while the host tolerance is vital to minimize tissue damage due to an excessive immune response. It was found that when challenged with the RNA virus, the host prevented overproduction of type I IFNs (interferons) to reduce host tissue damage, which validated the viewpoint at the molecular level [38]. It was also reported that the change in mean corresponded to the plasticity of the phenotype, whereas the change in variance reflected sensitivity to noise in tested environments [39]. In Fayoumi, we found smaller degrees in both the difference and fold change levels for both methylation as a sensor and regulator of the changes in the surrounding environment and gene expression as a functional response to the changes in the surrounding environment, even for housekeeping genes (Figure 5). We also observed that Fayoumi exhibited a clearer relationship between changes in methylation and gene expression compared to Leghorn chickens, indicating a more ordered or successful regulation or information transmission. Together with the observation of comparable virus titers in two infected breeds, this suggests a homeostasis hypothesis of avian influenza resistance in chickens as an indirect anti-virus mechanism, in addition to direct anti-virus mechanisms of destroying or blocking the entry of the virus into the cell (Figure 5). 

To validate that the obtained top-fold change genes were biologically meaningful, we analyzed the proteomics data of the sample samples (data not published). We found that most of the top-fold change genes with FDR-corrected *p* < 0.05 also showed significant differences at the protein level. Among the 85 (35+50) top-fold change genes, 17 transcripts could not be translated into proteins, and 12 top-fold change genes were labeled as uncharacterized proteins. We found that 18 top-fold change genes exhibited a significant difference in FA compared to FP and LA compared to LP. In order to clarify the change trends of top-fold change genes at the RNA and protein levels, we drew a heatmap with expression values measured as log_2_(expression values at the corresponding level of individuals with AIV inoculation/expression values of individuals challenged with PBS) (Appendix A). As shown in Appendix A, top-fold change genes obtained at the RNA level also showed the same regulation trend in the same comparison group.

Our homeostasis hypothesis for disease resistance in chickens can be regarded as an extension of the canalization hypothesis in the area of disease resistance. The canalization hypothesis predicted an invariable phenotype regardless of the environment variability [39]. In the process of AIV resistance, we suggest that genetics and epigenetics work together, contributing to the resistance phenotype. At the genetic differentiation level, we detected *MX1*, *HSP90AB1*, and *HSP90B1*, whose F_ST_ values passed the whole genome top 5% threshold. It is well known that non-synonymous codon changes could impact gene function. To identify functional sites within *MX1*, *HSP90AB1,* and *HSP90B1*, we compared dn/ds for these three genes. *MX1* is composed of 705 amino acids, and we detected that the 605th, 317th, 324th, and 354th amino acids were under positive selection with *p*-values of 0.03, 0.05, 0.05, and 0.05, respectively. The most interesting site in the *MX1* cDNA was Ser (S) to Asn (N) substitution at the amino acid position 631 (S631N) caused by mutation G2032A. However, whether this site contributes to AIV resistance is controversial. Previously, Wang et al. experimentally validated that the *MX1* amino acid 631 is not relevant to AIV replication in chickens both in ovo and in vivo [5]. In our analysis, the amino acid site 631 in *MX1* was not shown to be positively selected either. We applied the same method for *HSP90AB1* and *HSP90B1*; unfortunately, no significant site was found in *HSP90AB1* or *HSP90B1* with a *p*-value threshold of 0.1.

In our results, the MX1 protein was also shown to differ significantly for FA compared to FP and LA compared to LP in the lung tissues (Appendix A). After AIV inoculation, the Fayoumi breed upregulated the MX1 protein, whereas Leghorn chickens did not produce the MX1 protein (Appendix A). For the HSP90AB1 protein, in the Fayoumi breed, the fold change value of FA compared to FP was 5.1146 with a *p*-value of 2.3965 × 10^−111^. However, there was no significant difference for LA compared to LP (Appendix A). For the HSP90B1 protein, both breeds exhibited a significant difference after virus inoculation; however, Fayoumi upregulated HSP90B1 to a greater extent with a fold change of 42.3152, whereas the fold change of Leghorn chickens was 2.2874 (Appendix A). The *MX1* gene has been systematically studied in human and mice anti influenza virus infections. The *MX1* gene is a dynamin-like guanosine triphosphatase that blocks the importation of influenza ribonucleoprotein, a process known to be sufficient to confer resistance to influenza viruses [40,41]. In humans, the *MX1* gene product MxA inhibits a step of the influenza virus replication cycle following primary transcription [42]. Genetically modified *MX1+/+* mice showed more resistance to the influenza virus inoculation than *MX1*-deficient mice [43,44,45]. In our study, Fayoumi chickens showed a higher MX1 protein level after AIV inoculation, whereas expression of the MX1 protein remained at a relatively low level in Leghorn chickens both before and after inoculation. Our results suggested that the *MX1* gene may play an important role in the difference in AIV resistance between Fayoumi and Leghorn chickens. 

The HSP90 gene family is one of the best studied examples of a general capacitor that buffers both genetic and environmental perturbations [46]. In our result, we found that HSP90 showed high genetic differentiation level between the two breeds. In addition, the degrees of change and fold change levels in HSP90 in terms of methylation and gene expression were larger in Leghorn compared to Fayoumi chickens, which indicates higher robustness to AIV perturbations in Fayoumi, which is in line with the canalization hypothesis.

The key genes *HSP90AB1* and *HSP90B1* were reported to function in innate immunity, and we showed that these two HSP90 family genes also act as homeostasis markers in AIV infections. The *HSP90AB1* gene is one of the major two cytosolic isoforms of the HSP90 family [47]. The HSP90 gene family is relevant to numerous cellular pathways and is known to play roles in protein stabilization and the adaptive response to stress [48]. Wahl, A.; et al. found that approximately 30% of host changed proteins mediated by influenza directly interact with *HSP90AB1* [49]. By complexing with AKT, *HSP90AB1* activates downstream antiapoptotic NF-90FD5B20BF, which is an important aspect of cardiac cellular defense strategies [50]. By directly associating with *HMGB1*, *HSP90B1* plays an important role in the process of cellular and immune homeostasis at the airway mucosal surface [51]. Under this viewpoint, it is conceivable that increasing *HSP90AB1* and *HSP90B1* expression levels might efficiently reduce damage from AIV infections. For example, reactivating *HSP90AB1* expression by genetic modification or developing drugs to stimulate *HSP90AB1* expression might provide new opportunities for AIV resistance. In summary, evidence for genetic differentiation, methylation, and gene expression supports the hypothesis of maintaining homeostasis in disease resistance. Fayoumi chickens possess a superior ability to maintain homeostasis compared to Leghorn chickens, which could constitute another form of evolutionary robustness. However, further work is required to discover the specific gene functions that contribute to homeostasis in disease resistance in detail.

The avian influenza virus is one of the zoonotic pathogens that can cross the species barrier and result in epidemic outbreaks, which is in common with viruses in bats that can make the jump to humans. So far, multiple scholars have reported that bats remain disease-free when infected with the highly pathogenic RNA viruses they carry and bats can coexist with them without detectable fitness costs using measures such as overt signs of inflammation [52]. This is in line with findings from our results, which found that Fayoumi was more robust than Leghorn lines challenged with AIV. Based on the comparison of 10 bat genomes sequenced so far and mamal genomes, researchers found that PYHIN genes were entirely lost [53], which could interact with stimulator of interferon genes (STING) and activate the inflammasome. It has been hypothesized that the absence of the PYHIN family may allow bats to limit activation of the innate immune response to damaged self-DNA generated by RNA viral infection, thus avoiding excessive inflammation [53,54]. Zhou et al. found that if an antiviral immune pathway STING-interferon pathway in bats is dampened that bats can maintain just enough defenses against illness without triggering the immune systems from going into overdrive [55]. However, in humans and other mammals, an immune-based over-response to one of these and other pathogenic viruses can trigger severe illness. Dublin researchers have also shown that compared to the immune response of a mouse, bat macrophages can rapidly mount a robust antiviral response whenever a pathogen is detected, and, bat immune system can quickly reverse their response by releasing anti-inflammatory cytokines [56]. Moreover, bat MX1 significantly reduced the polymerase activity of viruses circulating in bats, including Ebola and influenza A-like viruses [57]. By increasing pathogen replication control and mitigating any immunopathology through decreased inflammatory responses, bat increased disease tolerance.

## Figures and Tables

**Figure 1 genes-10-00543-f001:**
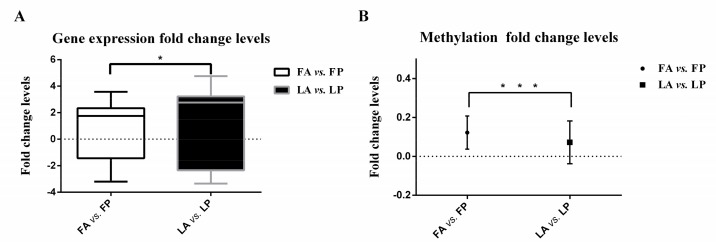
Gene and methylation expression fold change levels in Fayoumi and Leghorn breeds. (**A**) Gene expression fold change levels of top-fold change genes with false discovery rate (FDR)-corrected *p* < 0.05, which were measured as the variance of log_2_(gene expression levels of individuals with avian influenza virus (AIV) inoculation/gene expression levels of individuals challenged with phosphate buffer saline (PBS)). The comparison used F-tests, the y-axis represents mean with SD (Standard Deviation), and * indicates a significance level of 0.05; (**B**) methylation fold change levels of differentially methylated regions, which were measured as the variance of log_2_(methylation ratios of individuals with AIV inoculation/methylation ratios of individuals challenged with PBS). The comparison used F-tests, the y-axis represents the mean with SD, and *** indicates a significance level of 0.01.

**Figure 2 genes-10-00543-f002:**
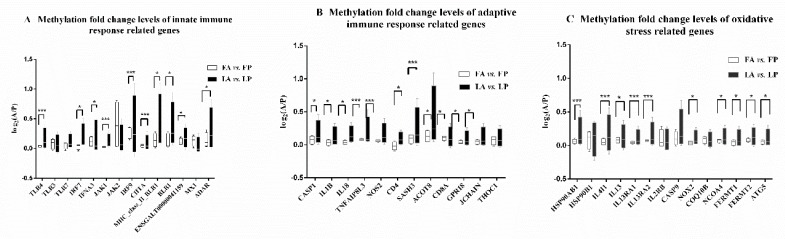
Gene methylation fold change levels of immune response related genes. The variance of log_2_(gene methylation ratios of individuals with AIV inoculation/gene methylation ratios of individuals challenged with PBS) were measured as the gene methylation fold change level. F-tests were used to compare gene methylation fold change levels in the Fayoumi and Leghorn breeds, * indicates a significance level of *p* = 0.01–0.05, *** indicates a significance level of *p* < 0.01. A,B,C represented innate immune response, adaptive immune response, and oxidative stress related pathways, respectively.

**Figure 3 genes-10-00543-f003:**
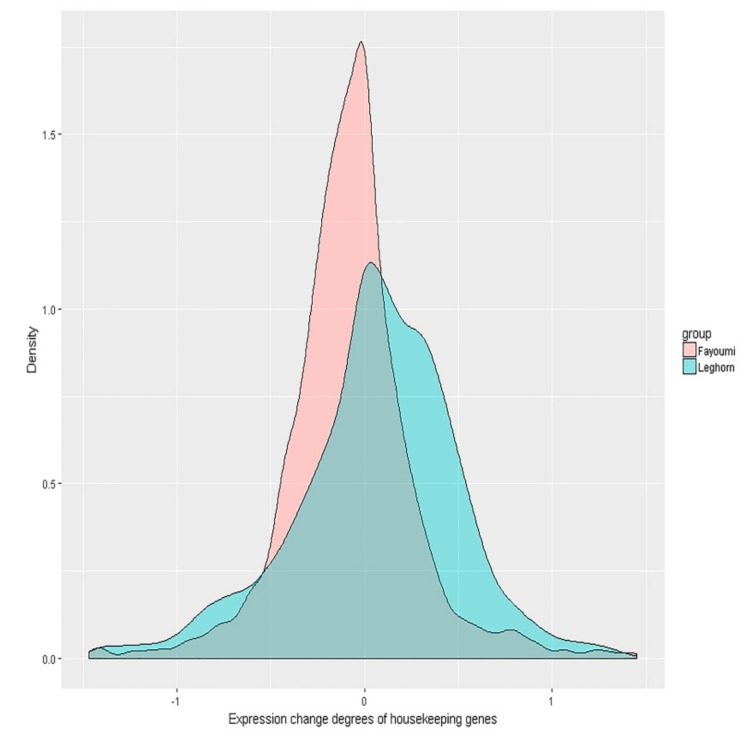
Housekeeping gene expression difference distribution in Fayoumi and Leghorn chickens. The x-axis represents the housekeeping gene expression difference for the two breeds. The values were calculated as log_2_(gene expression levels of Fayoumi individuals with AIV inoculation/gene expression levels of Fayoumi individuals challenged with PBS) or log_2_(gene expression levels of Leghorn individuals with AIV inoculation/gene expression levels of Leghorn individuals challenged with PBS). The y-axis represents the density of the gene expression difference degree.

**Figure 4 genes-10-00543-f004:**
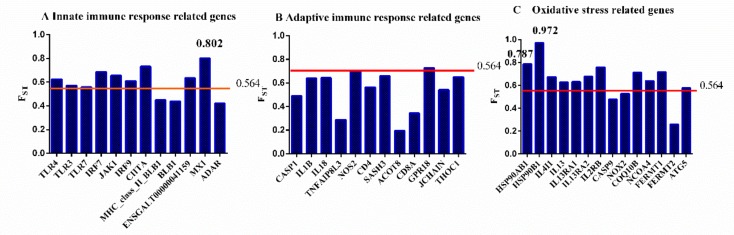
Fixation index (F_ST_) values of immune response related genes: (**A**) F_ST_ values of innate immune response related genes, (**B**) F_ST_ values of adaptive immune response related genes, (**C**) F_ST_ values of oxidative stress related genes. The red line represents the mean F_ST_ value of all genes in the genome.

**Figure 5 genes-10-00543-f005:**
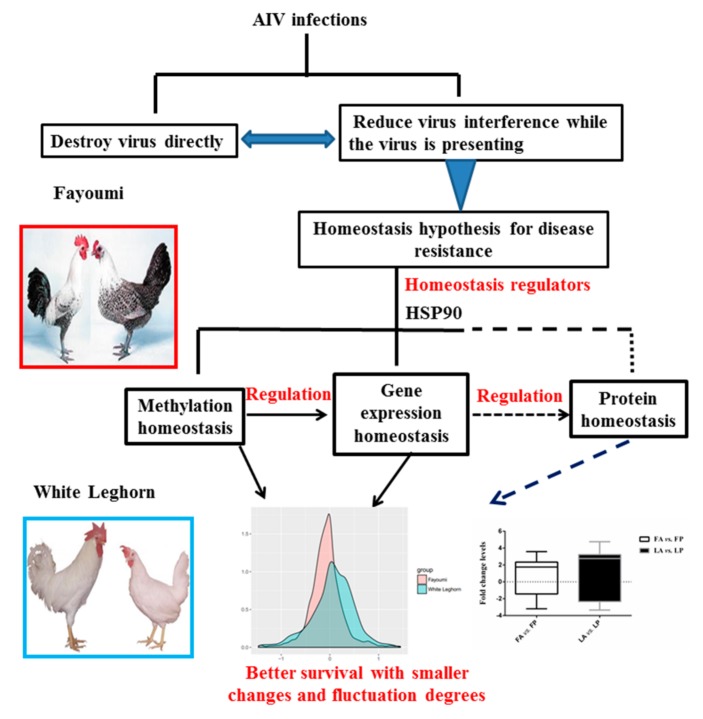
Scheme of the homeostasis hypothesis of avian influenza resistance in chickens. The overall homeostasis was achieved by methylation homeostasis, gene expression homeostasis, and protein homeostasis. The dashed line represents measurements that were not included in this study.

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
