# Peer review of "A Homeostasis Hypothesis of Avian Influenza Resistance in Chickens"

_genes, 2019, doi:10.3390/genes10070543_

Round 1

Reviewer 1 Report

The manuscript “A homeostasis hypothesis of avian influenza resistance in chickens” by An et al., compares the genomic, methylation and transcriptomic differences obtained from two chicken breeds “Leghorn GB2 and Fayoumi M43” in response to low pathogenic avian influenza virus infection of subtype H5N3. It is evident that the authors worked hard in different approaches to illustrate this difference. However, the manuscript can´t be accepted in its present form, and several points should be considered and addressed by the authors prior to being submitted for publication as detailed below:

Major:

-It is not clear for this reviewer whether the animal experiment conducted in this study is the same as performed by Wang et al., 2014 or just the same setup and repetition. For example: Wang et al., 2014 used 7 leghorns and 27 Fayoumi; in this study (line 93, eight lung tissues per line). Are they the same lung tissues used for RNA-seq? why not the same? Otherwise this study is focused on methylation only? Authors should make this point clearer for the reader.

Minor:

-Line 43: “Compared to human influenza, AIV has not been fully explored” there are thousands of publications on AIV, so better remove this sentence.

-Line 52-53: reference is missing

-Line 88: is it a new experiment or the same as conducted by Wang et al. 2014?

-Line 90: number of birds in each group?

-Line 98: “RNA-seq data analysis” as mentioned in line 177, not only the samples have been taken from Wang et al, but also the RNA-seq data. It is better to avoid long methodology and refer as described in Wang et al.

-178-180: “Eight three……..per treatment” repetition of material and methods.

-180-183: Same results as described by Wang et al., 2014.

Author Response

Major:

-It is not clear for this reviewer whether the animal experiment conducted in this study is the same as performed by Wang et al., 2014 or just the same setup and repetition. For example: Wang et al., 2014 used 7 leghorns and 27 Fayoumi; in this study (line 93, eight lung tissues per line). Are they the same lung tissues used for RNA-seq? why not the same? Otherwise this study is focused on methylation only? Authors should make this point clearer for the reader.

 Thanks for your comments. The RNA-seq data were generated from Wang et al. 2014, so they were exactly the same. As illustrated in the manuscript line 102-107,”Eight three-week-old chickens were randomly chosen from individuals challenged with a 107 50% egg infective dose of H5N3 AIV (LPAI) or PBS. There were two birds per breed per treatment”.

Minor:

-Line 43: “Compared to human influenza, AIV has not been fully explored” there are thousands of publications on AIV, so better remove this sentence.

Thank you for your suggestion. We have removed the sentence.

-Line 52-53: reference is missing

Thank you very much. We have added the reference and updated all the rests.

-Line 88: is it a new experiment or the same as conducted by Wang et al. 2014?

Thank you. RNA-seq data were generated from Wang et al. 2014 and methylation data were a new experiment.

-Line 90: number of birds in each group?

Thank you. We have added the relevant information.

-Line 98: “RNA-seq data analysis” as mentioned in line 177, not only the samples have been taken from Wang et al, but also the RNA-seq data. It is better to avoid long methodology and refer as described in Wang et al.

Thank you. We have shortened the methodology and cited the Wang et al.

-178-180: “Eight three……..per treatment” repetition of material and methods.

Thank you. We have deleted the sentences.

-180-183: Same results as described by Wang et al., 2014.

Thank you. We listed here to introduce our viewpoint.

Reviewer 2 Report

The paper by An and colleagues investigate the pathogenesis of avian influenza by comparing the transcriptome and epigenome of a highly susceptible chicken breed (Leghorn) to that of a relatively resistant breed (Fayoumi). The authors propose a homeostasis hypothesis whereby they speculate that Fayoumi chickens are relatively more resistant to disease due to their higher threshold of environmental perturbations (such as avian influenza). Overall the paper proposes an interesting hypothesis and the data add to the growing consensus that comparative genomics are useful mechanism to understand the pathogenesis of avian influenza. However, I have the following suggestions/queries to the authors to improve the manuscript:

1)      Can the authors please justify their choice of LPAI instead of HPAI? This seems surprising given that LPAI is asymptomatic in most chickens and the biggest breed-dependent differences in disease severity are observed after HPAI infection

2)      Can the authors please clarify the number of samples used per treatment group for RNASeq/epigenomics? They state that there were two birds per treatment. However, they also state two lung samples from the same breed and treatment group were pooled, making it sound as if there is only an n of 1 per treatment group. If this is the case how was a robust analysis (with statistical significance) performed?

3)      Can the authors please justify their use of Cuffdiff for transcriptomic analysis given that this method has a high rate of false positives

4)      Figure 1: It is difficult to understand how these treatment groups are significantly different given the overlapping error bars. Can the authors please adjust all figures in the manuscript so that the individual datapoints are shown (and overlayed on bars showing the mean +/- error) ?

5)      Figure 2 is very difficult to read, can the authors please enlarge/redo the figure so that it is legible

6)      The authors noted that HSP90AB1 is upregulated in Fayoumi chickens after infection – this is interpreted as enabling the chickens to better deal with environmental stressors. However, these data easily be interpreted as Fayoumi chickens mounting a greater stress response to influenza (hence the need for HSP90AB1 expression). Can the authors please comment on these two different interpretations

7)      The author’s central premise is that Fayoumi chickens can better maintain internal homeostasis following exposure to external stimulus. To truly support this hypothesis, it would be important to include additional experiments whereby the cells Fayoumi and Leghord chickens (potentially CEFs as these are easy to obtain) are challenged with a non-infectious external stimulous (e.g. high/low oxygen or temperature) and the same ‘homeostasis response’ is observed in Fayoumi cells

8)      The homeostasis hypothesis proposed herein is reminiscent of recent findings regarding bats and tolerance to a wide array to different range of pathogenic viruses. Can the authors please touch on this point in the discussion?

9)      The authors state “our results indicate that the MX1 gene plays a prominent role in the difference in AIV resistance between Fayoumi  and Leghorn chickens”. However, this is a vast overstatement of their findings, please modify accordingly.

Author Response

1) Can the authors please justify their choice of LPAI instead of HPAI? This seems surprising given that LPAI is asymptomatic in most chickens and the biggest breed-dependent differences in disease severity are observed after HPAI infection

Thank you very much. Avian influenza can be caused by highly pathogenic avian influenza (HPAI) virus or low pathogenicity avian influenza (LPAI). Compared with LPAI, HPAI virus infection could cause severe mortality up to 90% to 100% in chickens, within 48 hours. Whereas, poultry infected with LPAI only exhibit mild illness symptoms. LPAI is more likely to spread to other sheds due to a relatively low chance of detection and reporting compared to HPAI infection [1]. Moreover, numerous studies have investigated the specific mechanism of HPAI infection, while little is known about LPAI infection. Based on previous reports, LPAI also exhibited clinical signs in Fayoumi and Leghorn breed [2]. In the previous study, we did lesion severity tests in chicken lungs, which hit the peak at 4 days post-inoculation[2]. So we choose LPAI instead of HPAI.

2) Can the authors please clarify the number of samples used per treatment group for RNASeq/epigenomics? They state that there were two birds per treatment. However, they also state two lung samples from the same breed and treatment group were pooled, making it sound as if there is only an n of 1 per treatment group. If this is the case how was a robust analysis (with statistical significance) performed?

Thanks for your comments. For methylome study, there were 4 samples per treatment per line (2 biological samples * 2 technical repeats), so the statistical analyses were robust. For RNAseq study, we have to admit that experiments with biological replicates will, for sure, provide more accurate results. The expression data we used in this study came from a research from Prof. Huaijun Zhou’s group published in 2014. The experiment was conducted in 2010 actually. At that time, pooling RNA samples was common due to the high costs of RNA-sequencing. Although current DEG analysis methods encourage the use of biological replicates, there are reasonable statistical methods that handle data without replicates. For example, Chi-square/Fisher exact test method that counts reads for the target gene and reads for the rests. As an example, Trapnell et al., have compared expression data of consecutive time points, one sample for each time point. In their study, the cuffdiff software was used for DEG analysis without replicates [3].

Comparison of software packages for detecting differential expression in RNA-seq studies has shown that the number of DEGs plateaus at 4-6 replicates [4]. And the amount of biological replicates are required to be more from unrelated individuals than equivalent studies based on inbred ones [4]. Our experimental material Fayoumi and Leghorn are two inbred chicken lines whose inbreeding coefficients were over 99.9%. Since experiments without replicates are currently not preferred and will more likely introduce false positives, we chose more strict criteria for DEG to remove possible false positives, by setting the significant threshold as Benjamini-Hochberg-adjusted p values of 0.05. As shown in the manuscript, trends of change of DEGs at RNA and protein levels showed same pattern, which also validated our results.

3) Can the authors please justify their use of Cuffdiff for transcriptomic analysis given that this method has a high rate of false positives

Thank you for your comments. Cuffdiff can achieve similar performance compared with DESeq when the sequencing depth is more than 20 million reads for each individual sample [5]. Cuffdiff transformed the alignment results to FPKMM, which reduced sample variability compared with raw counts [6]. Compared with other DEGs identification tools, edgeR introduced the most false positives compared with cuffdiff and DESeq [5]. As shown in the manuscript, trends of change of DEGs at RNA and protein levels showed same pattern, which also validated our results.

4) Figure 1: It is difficult to understand how these treatment groups are significantly different given the overlapping error bars. Can the authors please adjust all figures in the manuscript so that the individual datapoints are shown (and overlayed on bars showing the mean +/- error) ?

Thank you for your advice. We have already redrawn figure1 and adjusted figure2 in order that figures in the manuscript could clearly show different trends in treatment groups.

5) Figure 2 is very difficult to read, can the authors please enlarge/redo the figure so that it is legible

Thank you for your suggestion. We have already enlarged and rotated the figure 2 so that readers could know our meaning clearly.

6) The authors noted that HSP90AB1 is upregulated in Fayoumi chickens after infection – this is interpreted as enabling the chickens to better deal with environmental stressors. However, these data easily be interpreted as Fayoumi chickens mounting a greater stress response to influenza (hence the need for HSP90AB1 expression). Can the authors please comment on these two different interpretations

Thank you for your advice. HSP90AB1 is one of the major two cytosolic isoforms of the HSP90 family [7]. HSP90 is relevant to numerous cellular pathways and is known to play roles in protein stabilization and the adaptive response to stress [8]. Wahl, A et al. found that approximately 30% of host changed proteins mediated by influenza directly interact with HSP90AB1[9]. By complexing with AKT, HSP90AB1 activates downstream antiapoptotic NF-90FD5B20BFwhich is an important aspect of cardiac cellular defense strategies [10]. Under this viewpoint, it is conceivable that increasing HSP90AB1 and HSP90B1 expression levels might efficiently reduce damage from AIV infections. For example, reactivating HSP90AB1 expression by genetic modification or developing drugs to stimulate HSP90AB1 expression might provide new opportunities for AIV resistance. So our viewpoint is that increased HSP90AB1 expression could reduce tissue damage.

7) The author’s central premise is that Fayoumi chickens can better maintain internal homeostasis following exposure to external stimulus. To truly support this hypothesis, it would be important to include additional experiments whereby the cells Fayoumi and Leghord chickens (potentially CEFs as these are easy to obtain) are challenged with a non-infectious external stimulous (e.g. high/low oxygen or temperature) and the same ‘homeostasis response’ is observed in Fayoumi cells

Thank you for your suggestion. To provide further evidence of the homeostasis hypothesis of disease resistance, we reanalyzed the RNA-seq data generated by Deist et al., 2017. In their study, they did Newcastle disease virus (NDV) challenging experiments using inbred Fayoumi (M 15.2) and Leghorn (GH 6.) lines [11]. A total of 48 RNA-seq dataset they generated came from lung samples at 2 days, 6 days and 10 days post infection, for both challenged and nonchallenged individuals [11]. To test our hypothesis, the differential expression analyses were conducted in the same way as we performed in the AIV manuscript. DEGs were obtained at the significance level of p < 0.05 (adjusted by FDR), which was the same as our AIV gene expression method. By using the same gene expression fluctuation definition, gene expression fluctuation pattern between two chicken lines at three time points were shown in the following figure. For the whole process of three time points, Fayoumi gene expression fluctuation degree was obviously smaller than Leghorn gene expression fluctuation degree. At 6 days post infection, Fayoumi has no DEGs, which was also obtained by Deist et al., 2017 [11]. It is clear that, compared to the trend of White Leghorn, Fayoumi exhibited less gene expression fluctuations after challenging, which verified our hypothesis.

For each time point, the NDV fluctuation degree was measured as the mean value of log2(gene expression levels of individuals injected with NDV/ gene expression levels of individuals injected with mock) for DEGs, with a threshold setting to p < 0.05 (adjusted by FDR).

Besides, resistance to Salmonella enteritidis also studied in heterophils from Leghorn and Fayoumi chickens, and compared to Leghorn lines, Fayoumi favored reduced or suppressed inflammatory response, which was in line with our hypothesis [12]. We have also searched data challenged with a non-infectious external stimulous in Fayoumi and Leghorn lines, whereas there were no suitable data available. Due to the limitation that our lab focus on data analysis, we could not get Fayoumi and Leghorn chicken cell samples immediately, we will further validate the hypothesis challenged with a non-infectious external stimulous in the next step.

8) The homeostasis hypothesis proposed herein is reminiscent of recent findings regarding bats and tolerance to a wide array to different range of pathogenic viruses. Can the authors please touch on this point in the discussion?

Thank you for your suggestion.

Avian influenza virus as one of the zoonotic pathogens that can cross the species barrier and result in epidemic outbreaks, which has in common with viruses in bats that can make the jump to humans. So far, multiple scholars have reported that bats remain disease-free when infected with the highly pathogenic RNA viruses they carry and bats can coexist with them without detectable fitness costs using measures such as overt signs of inflammation [13]. This is in line with findings from our results, which found that Fayoumi was more robust than Leghorn lines challenged with AIV. Based on comparison of 10 bat genomes sequenced so far and mamal genomes, researchers found that PYHIN genes were entirely lost [14], which could interact with stimulator of interferon genes (STING) and activate the inflammasome. It has been hypothesized that the absence of the PYHIN family may allow bats to limit activation of the innate immune response to damaged self-DNA generated by RNA viral infection, thus avoiding excessive inflammation [14, 15]. Zhou et. al found that if an antiviral immune pathway STING-interferon pathway in bats is dampened that bats can maintain just enough defenses against illness without triggering the immune systems from going into overdrive [16]. However, in humans and other mammals, an immune-based over-response to one of these and other pathogenic viruses can trigger severe illness. Dublin researchers have also shown that compared to the immune response of a mouse, bat macrophages can rapidly mount a robust antiviral response whenever a pathogen is detected, and, bat immune system can quickly reverse their response by releasing anti-inflammatory cytokines [17]. Moreover, bat Mx1 significantly reduced the polymerase activity of viruses circulating in bats, including Ebola and influenza A-like viruses [18]. By increasing pathogen replication control and mitigating any immunopathology through decreased inflammatory responses, bat increased disease tolerance.

9) The authors state “our results indicate that the MX1 gene plays a prominent role in the difference in AIV resistance between Fayoumi  and Leghorn chickens”. However, this is a vast overstatement of their findings, please modify accordingly.

Thank you for your advice. We have corrected the sentence as “Our results suggested that the MX1 gene may play an important role in the difference in AIV resistance between Fayoumi and Leghorn chickens. “

References:

 1.    Angela Bullanday Scott J, Jenny-Ann L. M. L. Toribio, Mini Singh, Groves P, Belinda Barnes KGBM, And Marta Hernandez-Jover: Low- and high-Pathogenic avian influenza H5 and H7 spread risk assessment within and between Australian commercial chicken Farms. Frontiers in Veterinary Science 2018, 5:63.

 2.    Wang Y, Lupiani B, Reddy SM, Lamont SJ, Zhou H: RNA-seq analysis revealed novel genes and signaling pathway associated with disease resistance to avian influenza virus infection in chickens. POULTRY SCI 2014, 93(2):485-493.

 3.    Trapnell C, Williams BA, Pertea G, Mortazavi A, Kwan G, van Baren MJ, Salzberg SL, Wold BJ, Pachter L: Transcript assembly and quantification by RNA-Seq reveals unannotated transcripts and isoform switching during cell differentiation. NAT BIOTECHNOL 2010, 28(5):511-515.

 4.    Seyednasrollah F, Laiho A, Elo LL: Comparison of software packages for detecting differential expression in RNA-seq studies. BRIEF BIOINFORM 2015, 16(1):59-70.

 5.    Zhang ZH, Jhaveri DJ, Marshall VM, Bauer DC, Edson J, Narayanan RK, Robinson GJ, Lundberg AE, Bartlett PF, Wray NR et al: A comparative study of techniques for differential expression analysis on RNA-Seq data. PLOS ONE 2014, 9(8

):e103207.

 6.    Giorgi F M DFCL: Comparative study of RNA-seq- and Microarray-derived coexpression networks in Arabidopsis thaliana. BIOINFORMATICS 2013, 6(29):717-724.

 7.    Chen B, Piel WH, Gui L, Bruford E, Monteiro A: The HSP90 family of genes in the human genome: Insights into their divergence and evolution. GENOMICS 2005, 86(6):627-637.

 8.    Makhnevych T, Houry WA: The role of Hsp90 in protein complex assembly. Biochimica et Biophysica Acta (BBA) - Molecular Cell Research 2012, 1823(3):674-682.

 9.    Wahl A, Schafer F, Bardet W, Hildebrand WH: HLA class I molecules reflect an altered host proteome after influenza virus infection. HUM IMMUNOL 2010, 71(1):14-22.

10.    Grabek KR, Karimpour-Fard A, Epperson LE, Hindle A, Hunter LE, Martin SL: Multistate proteomics analysis reveals novel strategies used by a hibernator to precondition the heart and conserve ATP for winter heterothermy. PHYSIOL GENOMICS 2011, 43(22):1263-1275.

11.    Deist MS, Gallardo RA, Bunn DA, Dekkers JCM, Zhou H, Lamont SJ: Resistant and susceptible chicken lines show distinctive responses to Newcastle disease virus infection in the lung transcriptome. BMC GENOMICS 2017, 18(1).

12.    Redmond SB, Chuammitri P, Andreasen CB, Palić D, Lamont SJ: Chicken heterophils from commercially selected and non-selected genetic lines express cytokines differently after in vitro exposure to Salmonella enteritidis. VET IMMUNOL IMMUNOP 2009, 132(2-4):129-134.

13.    Baker ML, Schountz T, Wang LF: Antiviral Immune Responses of Bats: A Review. ZOONOSES PUBLIC HLTH 2013, 60(1):104-116.

14.    Ahn M, Cui J, Irving AT, Wang L: Unique Loss of the PYHIN Gene Family in Bats Amongst Mammals: Implications for Inflammasome Sensing. SCI REP-UK 2016, 6(1).

15.    Li N, Parrish M, Chan TK, Yin L, Rai P, Yoshiyuki Y, Abolhassani N, Tan KB, Kiraly O, Chow VTK et al: Influenza infection induces host DNA damage and dynamic DNA damage responses during tissue regeneration. CELL MOL LIFE SCI 2015, 72(15):2973-2988.

16.    Xie J, Li Y, Shen X, Goh G, Zhu Y, Cui J, Wang L, Shi Z, Zhou P: Dampened STING-Dependent Interferon Activation in Bats. CELL HOST MICROBE 2018, 23(3):297-301.

17.    Joanna Kacprzyk, Graham M. Hughes, Eva M. Palsson-McDermott, Susan R. Quinn, J. S, Puechmaille, Luke A. J. O'Neill, Teeling EC: A Potent Anti-Inflammatory Response in Bat Macrophages May Be Linked to Extended Longevity and Viral  Tolerance. Acta Chiropterologica 2017, 2(19):219-228.

18.    Jonas Fuchs, Martin Hölzer, Mirjam Schilling, Corinna Patzina, Andreas Schoen, Thomas Hoenen, Gert Zimmer, Manja Marz, Friedemann Weber, Marcel A. Müller et al: Evolution and antiviral specificity of interferon-induced Mx proteins of bats against Ebola-, Influenza-, and other RNA viruses. J VIROL 2017, 91(15):e317-e361.

Round 2

Reviewer 1 Report

Authors have addressed all comment qnd agree with the current version

Reviewer 2 Report

Accept